# Science Teaching at a Distance in Greece: Students' Views

**Constantina Stefanidou** * and **Achilleas Mandrikas**

Athens Science and Education Laboratory, Department of Primary Education, School of Education,
National and Kapodistrian University of Athens, 10676 Athens, Greece
* Correspondence: sconstant@primedu.uoa.gr

**Abstract:** This article presents in a comparative way the findings from two surveys conducted on primary (students in Y6) and secondary (students in Y10) students in Attica, Greece, in order to map their views on distance science education, which was emergently implemented due to the COVID-19 outbreak during the school year 2020–2021. The research was conducted in a sample of 378 primary and 197 secondary students. The findings revealed that students were not satisfied with the distance teaching and learning of science, either in primary or in secondary education, except for the increased use of audiovisual material. Technical issues, such as poor network and infrastructure, lack of face-to-face interaction with classmates and teacher, external and internal distractions, lack of sufficient experimental activities, and limited understanding of concepts were common findings in both levels of education. Concerning the differences between the levels, it seems that in primary education more technical problems were reported, while in secondary education more didactic problems were reported.

**Keywords:** science education; science teaching at a distance; students' views; COVID-19 pandemic

## 1. Introduction

The COVID-19 pandemic affected education at all levels worldwide. Since the COVID-19 outbreak in February 2020, 191 countries changed rapidly and mainly with no planning from the face-to-face learning process to distance learning at all levels of education [1,2]. Greece is one of the countries in which schools were closed for a very extended period both in primary and secondary education [3].

This suspension was accompanied by a transition from in-person to emergency and unprepared remote education, combining synchronous and asynchronous distance learning. Even though distance learning enabled education to stay alive at all levels, at the same time several challenges emerged including internet access, lack of infrastructure, limited or no teachers' training, students' engagement, and the results of the learning process. The COVID-19 pandemic required a rapid transition which was a great challenge for both teachers and students.

In this paper, the views of students in Y6 and Y10 on distance science teaching during the COVID-19 pandemic in Greece are investigated. We addressed students in Y6 because sixth grade in Greece is the last primary education grade and students in Y10 because they have a lot of experience of science classes and are not yet affected by university exam choices. This paper's topic is driven by the fact that distance teaching was implemented worldwide without a deep exploration of students' attitudes, views, and performance, especially in a subject like science which is considered to include a remarkable laboratory component.

## 2. Literature Review

"Distance education" is meant to include synchronous and asynchronous educational environments which presuppose the extended use of technological means, such as efficient

internet services and personal devices, along with experienced teachers and specially designed teaching and learning sequences, in order for the lack of physical communication to be moderated. "Online learning" offers real time communication and interaction, although students and teachers are connected from different places, usually their homes. A digital platform is usually the educational environment, in which teachers develop their teaching activities and call students for participation. Appropriate teaching practices, along with adequate infrastructure, enhances students' participation and makes online education effective [4].

The literature on distance education is very rich. Students' engagement in the learning process and interaction between each other is a component highly demonstrated by many researchers [5–8]. Students' active involvement, relationships between students, relationships between students and teachers, and students' motivation are often underlined as important components of effective distance learning environments [4,8,9].

In the last two years, there is an increased interest in science education research concerning the implementation conditions of distance learning due to the COVID-19 pandemic. Some research revealed problems with internet access and lack of infrastructure [1,10–13], others revealed the importance of the conditions existing in home environment [14–16], others recorded the difficulties faced by students in terms of communication, self-discipline, and motivation [17–19], and others examined students' readiness to learn online and also teachers' readiness to teach online [20–22]. Moreover, an increase of inequalities in learning opportunities during the COVID-19 pandemic was recorded by several researchers [2,23–25].

OECD examined the responses of 98 countries to the disruption of school-based education for several months. In the report, several aspects of online education were investigated, such as schools' and teachers' preparedness, the adequacy and availability of technology, students' access to the digital world, and students' access to a quiet place in their homes [22]. The report indicated that several countries like Greece suffer from lack of most prerequisites that could support science distance education. For example, in Greece, the percentage of students in schools whose principal agreed that the school has sufficient qualified technical assistant staff is the second lowest among 98 countries. Moreover, on average across OECD countries, there is almost one computer available at school for every 15-year-old student for educational purposes. In Greece, the corresponding ratio is just 0.25, which means that there is only one computer for every four students.

A similar report coming from the European Union sounds alarm on both the short- and long-term consequences of COVID-19 in education. According to the report, the educational loss by the change from school-based to remote schooling education is taken for granted. Educational institutions all over the world have acknowledged problems such as lack of student–student and student–teacher communication, increased stress, and lack of motivation [1].

In Greece, Anastasiades [26] described in detail the implementation of distance education during the COVID-19 pandemic emphasizing the pedagogical dimension and the opportunity "of transition to the open school of inquiry-based learning". Karadimou and Tsioumis [27] described the general impact of the COVID-19 pandemic on the Greek educational community and focused on official instructions derived from the Ministry of Education. Geropoulos et al. [28] have taken 43 secondary head teachers' interviews and showed "the inadequate level of readiness of the state mechanism to cope with the educational requirements arising from the crisis" (p. 60).

However, research on the distance teaching of specific teaching subjects during the COVID-19 pandemic is relatively limited. A few studies concerning primary and secondary education in various countries have tried to reveal the features of science teaching during the COVID-19 pandemic.

In the USA, Macias et al. [29] conducted a study on the impact of the COVID-19 pandemic on science teaching and science teachers of the eighth grade. The answers of 515 teachers ascertained much less student engagement in remote learning, less time spent

on science lessons, and little implementation of the instructional methods aligned with NGSS like investigations, group work, or analyzing data. In contrast, watching videos and using online simulations and reading material emerge as the most used practices (p. 4).

In Canada, McPherson and Pearce [30] investigated tensions and contradictions of practices that science teachers faced during the COVID-19 pandemic through professional development of ten secondary science teachers. Teachers reported "frustrations with student engagement during online lessons" (p. 8) attributing this behavior to less opportunities to interact with experiments, demonstrations, or modeling. Moreover, teachers struggled with students' fair evaluation and with the need to shift their own professional practices.

In Indonesia, Wisanti et al. [12] recorded 177 secondary science teachers' difficulties and challenges regarding science teaching during the COVID-19 pandemic. Findings concerned three main factors: technology, students, and teachers. Concerning technology, most teachers (77.5%) had difficulties with management of online learning especially with internet access. Students showed low motivation, lack of discipline, and lack of communication equipment. The greatest challenge that teachers faced was concept explanation without practical work implementation. Half of the teachers tried to change practical work with another task, 20% skipped it completely, and only 3% used a virtual laboratory to replace practical work. Moreover, a significant 36% declared a lack of application operating skills (p. 8).

In UAE, Al Darayseh [31] investigated the impact of the COVID-19 pandemic on the modes of teaching of 62 secondary science teachers. According to 46% of the surveyed teachers, the main challenge of the online learning environment used during the pandemic was the lack of hands-on activities and experiments. Another problem was limited interaction in the online classroom accompanied by teachers' difficulties with fostering interaction between students and teachers. The management of students' behavior and the management of technical issues were recorded to a lesser extent. These findings probably explain teachers' preference to teach science via a traditional classroom (46%), while 32% would prefer blended teaching, and 22% virtual teaching (p. 114).

In Turkey, Avsar Erumit et al. [32] explored 37 science teachers of fifth through eighth grade and their adaptations to online teaching. Teachers found it very difficult to motivate their students; this is the reason they looked for fun ways to keep students engaged as well as interactive tools "such as web 2.0 tools, virtual labs, interactive games, and videos" (p. 44). In another study, six elementary teachers' transition to remote teaching was explored with emphasis on inquiry-based teaching practices [33]. Teachers admitted that the nature of the activities in remote teaching "did not match inquiry-based instruction" (p. 74), even if they used models or online videos. In contrast, when parents were asked to judge remote teaching, they found that "videos used during science instruction were arresting" [34] (p. 1902), but they complained about insufficient corrective feedback, short lesson duration, the absence of experiments, and the lack of students' active participation and social interaction between students (p. 1906).

In the UK, Leonardi et al. [35] have taken in mind the impact of the COVID-19 pandemic on primary science education in the annual report of the Wellcome Trust CFE Research 2021. Via this project, data on the way science is taught across the UK, including time spent and attitudes towards science, are annually reported. In 2021, a sample of 2823 primary teachers focused on the use of Explorify, a free digital resource for science teaching. The majority of teachers agree that science teaching was affected by the pandemic, meaning that less science was taught remotely, the curriculum was not fully covered, they couldn't work scientifically, and more adaptations than in other subjects were needed to teach science at a distance [35] (p. 3). In particular, teachers declared that they have taught less science (56%) and have used fewer investigations (80%) (p. 22). Gaps in students' knowledge and investigation skills were reported by 67% of teachers (p. 38). Teachers found it difficult to differentiate science lessons (82%) and felt unable to assess students' progress (62%) (p. 23). Moreover, fundamental science activities, such as observation, recording, and analyzing data decreased in remote teaching (p. 40), while sharing videos in

online class and at home increased (p. 46). In conclusion, 72% of teachers find it difficult to teach science remotely and 54% agree that is easier to teach other subjects remotely (p. 23).

In Greece, Stefanidou et al. [36] investigated secondary students' views on distance physics teaching during the COVID-19 pandemic revealing positive and negative aspects of online education. In a similar way, Mandrikas et al. [37] investigated primary students' views on distance science teaching during the COVID-19 pandemic demonstrating several problems and poor knowledge results. Both studies highlight students' views on distance teaching and learning, which is very important as students are the receivers of any change made in education during the COVID-19 pandemic. Moreover, they refer to science, a particular subject with laboratory requirements for in-class teaching.

Nevertheless, research on the differences between science distance teaching in primary and secondary education in the same country is missing. In the present study, the views of students in Y6 and Y10 on science distance teaching in Greece are compared, so as to reach conclusions concerning science distance teaching during the COVID-19 pandemic throughout the system of compulsory education in Greece. We consider this overview interesting, because it could provide the possibility of locating deeper insights about science teaching regardless of the pandemic. Genuine interest, provision of experimental equipment, teaching methods, science practices, time spent, amount of knowledge provided, type of skills cultivated, and students' response to science teaching are some of the issues that determine the reception of science in school at any educational level. Therefore, emerging similarities, differences, discontinuities, difficulties, requirements, and recommendations could be useful to science teachers, stakeholders, and science curriculum designers. Moreover, according to the findings, some practical implications depending on the age of the students could be suggested for more effective science teaching in any teaching circumstances. Finally, the results could be compared with those of other countries and contribute to a further discussion on science education.

## 3. Methodology

### 3.1. Research Question

The research question of this study is as follows: What are primary and secondary Greek students' views about the distance teaching of science during the COVID-19 pandemic regarding (a) students' interest in science, (b) students' communication with teachers and classmates, (c) changes in teaching practices, (d) concentration and understanding of science concepts, and (e) students' overall evaluation of science teaching at a distance.

### 3.2. Sample

The sample of the present study consists of 378 primary students (students in Y6, 12 years old, 165 male and 213 female) and 197 secondary students (students in Y10, 16 years old, 80 male and 117 female), who voluntarily participated in the research. These students attended public schools in Attica (Greece), 24 elementary and 20 secondary schools, which derived from all district areas of Athens and Piraeus representing every social and economic background.

As part of the national strategy for the confrontation of the pandemic, all students in Greece attended distance teaching during almost the last two school years. Specifically, during the previous year (2019–2020) schools were normally open from September to March, when the World Health Organization (WHO) declared COVID-19 a pandemic. Therefore, students continued science education remotely until June and in-person again for just 15 days before the end of the school year in June 2020. Regarding the next school year (2020–2021), schools opened for just one month at the beginning of the school period and went to distance education for the rest of the school year.

### 3.3. Data Collection

Two similar questionnaires, appropriate for each educational stage, were created and used for data collection. The questionnaires consisted of twenty-three (23) Likert scale,

either 1–5 or 1–3, closed-ended questions and two (2) open-ended questions to shed light to the qualitative characteristics, and particularly to map students' views on what they liked most and least in science teaching by distance. The questions were organized in five categories, according to the previously mentioned aspects which the research question is analyzing. These categories were formed based on corresponding classifications in similar studies [1,5,8,20,22].

The validity of the research tools was supported due to the correspondence of the questions' content to students' views on distance education (content validity) and vice versa, the questionnaires included all aspects of science teaching at a distance, as they were organized in the five groups mentioned above. Moreover, the questionnaires were thoroughly tested by two expert, experienced teachers at each stage of education, to adjust the language in primary and secondary students' levels. Clear instructions and explanations were given to students in order to complete the questionnaire.

The questionnaires were transformed in digital form and distributed electronically, attached to a cover letter for parents and students. Finally, they were answered anonymously outside of school hours.

### 3.4. Data Analysis

Closed-ended questions were categorized according to the pre-defined response grades and are presented in tables.

For the analysis of open-ended questions, an inductive approach for qualitative data analysis was used [38]. Students' answers were indexed and categorized according to their content by the first author who grouped similar answers creating codes. The second author repeated the inductive process. The level of agreement between the two authors (coders) was depicted to Cohen's kappa 0.8. A third coder helped with the points of disagreement. Descriptive statistics were used to quantify the findings and provide a clearer picture of the similarities and differences between primary and secondary students' views on distance science education. An independent t-test was used to determine whether there was a statistically significant difference between the results of the two groups. An alpha level of 0.05 for all statistical tests was used.

Quality criteria for supporting trustworthiness were implemented. Credibility was established by peer debriefing. Particularly, an expert in the field thoroughly explored the methodology followed, and the data acquisition and analysis, as well as the formation of the findings. He served as a critical reviewer, asking questions and making recommendations. Transferability is enhanced by the fact that the authors describe in detail the research, from its context to its results, informing interested researchers about the possible repeatability to further situations. What follows is a comparative presentation of the findings of the two different groups (primary and secondary students).

## 4. Results and Discussion

### 4.1. Regarding Students' Interest in Science

Students were asked about their interest in science. Table 1 shows that in both educational levels (primary and secondary education) more than half of the students had increased interest in science. In primary education, students who found science interesting and very interesting are 83.5% which is a much bigger percentage than the corresponding percentage of secondary education, which is 50.2%. Primary education students seem to find science classes more interesting (M = 4.16, Mdn = 4, SD = 0.77) than secondary education students (M = 3.49, Mdn = 3.33, SD = 0.89), $t(576) = 1.96$, $p < 0.05$. Younger students seem to have an innate curiosity and willingness to explore science topics, which explain how the planet, ecosystems, and living organisms work, and interpret everyday life phenomena. However, with the passage of years and schooling this interest declines and it is important to determine the causes of this decline.

**Table 1.** Students' interest in science.

|  | Not Interesting | Slightly Interesting | Moderately Interesting | Interesting | Very Interesting |
|---|---|---|---|---|---|
| Primary Education | 0.3% | 2.4% | 13.8% | 47.9% | 35.6% |
| Secondary Education | 5.1% | 11.7% | 33.0% | 29.2% | 21.0% |

*4.2. Regarding Teacher–Student and Student–Student Communication*

Table 2 includes students' answers regarding teacher–student communication in primary and secondary education during the pandemic. Almost half of the primary education students answered that the communication between teachers and students became worse while almost half of secondary students considered the communication to remain the same, as in face-to-face teaching. Primary students considered the teacher–student communication to be much worse (M = 1.44, Mdn = 1, SD = 0.50) than secondary students (M = 1.76, Mdn = 2, SD = 0.57), $t(576) = 1.96$, $p < 0.05$. The difference between the educational levels probably indicates that student–teacher communication in primary education is better than in secondary education, which is why students felt that they were deprived of this important component of teaching during the science distance teaching.

**Table 2.** Teacher–student communication.

|  | Became Worse | Remained the Same | Improved |
|---|---|---|---|
| Primary education | 48.7% | 43.4% | 7.9% |
| Secondary education | 37.9% | 48.4% | 13.7% |

Regarding the communication between students, the difference in findings between primary and secondary education are even more pronounced. The majority of primary education students (67.2%) considered that the communication between classmates grew worse, while the corresponding percentage of secondary education students is only 33.3%. Most of the secondary students (53.3%) answered that the communication with their classmates remained the same (Table 3), while the percentage of primary students that considered the interaction with classmates improved was half of the corresponding percentage of secondary education students. Primary education students found the communication between classmates more problematic (M = 1.26, Mdn = 1, SD = 0.44) than secondary education students (M = 1.78, Mdn = 2, SD = 0.61), $t(576) = 1.96$, $p < 0.05$. These findings probably indicate that peer-to-peer communication is at higher levels in primary education than in secondary education, which is why primary students felt the reduction imposed due to the pandemic.

**Table 3.** Student–student communication.

|  | Became Worse | Remained the Same | Improved |
|---|---|---|---|
| Primary education | 67.2% | 26.2% | 6.6% |
| Secondary education | 33.3% | 53.3% | 13.4% |

Limited communication between students and between teachers and students has also been recorded by other researchers of science teaching during the COVID-19 pandemic [31,35]. Students' engagement in remote teaching in comparison with in-person teaching has been recorded as "much less" by Macias et al. [29] while McPherson and Pearce [30] concluded that "teachers struggled with a lack of student engagement" (p. 8).

*4.3. Regarding Teaching Practices*

Students were asked whether they observed changes in teaching methods, practices, and the assigned homework during the science distance teaching due to the COVID-19 pandemic. In Table 4, students' views on differences between teachers' practices in in-person and distance teaching are presented. In both educational levels, most students answered that there were changes in science teaching practices during the pandemic period, namely, 60.7% of primary education students and 51.8% of secondary education students. Primary students observed more changes in teaching practices (M = 3, Mdn = 3, SD = 1.20) than secondary education students (M = 2.70, Mdn = 2.67, SD = 0.99), $t(576) = 1.96$, $p < 0.05$. This means that students believe that teachers of both educational levels tried to adapt to the new teaching conditions to a certain degree with slightly higher percentages in primary education. According to students' answers, only a limited percentage of teachers did not make changes in teaching practices, probably due to poor familiarity with the needed digital technology or due to very limited access to infrastructure.

**Table 4.** To what extent did teachers change their teaching practices during their science teaching at a distance?

|  | Not at All | Little | Moderately | Much | Very Much |
|---|---|---|---|---|---|
| Primary education | 14.4% | 24.9% | 24.4% | 24.9% | 11.4% |
| Secondary education | 14.7% | 33.5% | 29.6% | 11.7% | 10.5% |

Adaptations of teaching practices were considered necessary by teachers in all relevant surveys. Teachers had to adapt their teaching pace when implementing remote learning [35]. Many teachers adapted teaching material in order to engage their students in some type of inquiry, for example they used science videos more often than in in-person classes [33]. In general, changes of teaching tools were necessary to shift from face-to-face experimenting to online experimenting [29–32]. Even parents clearly could notice that "practices as experiments were not implemented in distance education" [34] (p. 1906).

Table 5 shows the amount of students' workload during science distance teaching in comparison to in-person teaching. Almost half of the students in both stages considered that the workload remained the same, and a fair percentage of them (39.2% in primary and 32.3% in secondary education, if "much decreased" and "decreased" is added), considered that workload demands decreased. In primary education (M = 2.67, Mdn = 3, SD = 0.84), homework seems to have decreased slightly more than in secondary education (M = 2.88, Mdn = 3, SD = 0.91), $t(576) = 1.96$, $p < 0.05$, possibly due to the younger age of the students. Also, in secondary education the trends of increasing homework were slightly higher (22%) compared to primary education (15%), which can be attributed to teachers' worry about covering the curriculum before the exam period.

**Table 5.** Workload in the context of science teaching at a distance.

|  | Much Decreased | Decreased | Remained the Same | Increased | Much Increased |
|---|---|---|---|---|---|
| Primary education | 9.0% | 30.2% | 46.0% | 12.7% | 2.1% |
| Secondary education | 10.5% | 21.8% | 45.2% | 15.1% | 7.4% |

A possible explanation for the decreased workload could be the fact that teachers spent overall less time with students and thus spent less time on science [29], (p. 4). In one case, parents noticed that "corrective feedback was insufficient" [34], (p. 1906), which also implies a decrease in the workload assigned by teachers. Another reason for the decreased workload could be teachers' difficulty in providing "accurate and fair evaluations" [30], (p. 10), a point also reported by other researchers [12].

### 4.4. Regarding Students' Concentration and Conceptualization

In the fourth group of items, students were asked about their concentration during the lesson (Table 6) and the conceptualization of new concepts (Table 7). Students' responses revealed that most of them suffered from lack of concentration during remote teaching and they stated that they had poor understanding of the lesson content. Table 6 reveals that secondary education students was less able to concentrate (58.4%) than primary students (42.0%). Even the students who estimated that they could concentrate in remote schooling as well as in face-to-face conditions were clearly more in primary (48%) than in secondary education (29%), $t(576) = 1.96$, $p < 0.05$. It seems that in adolescence the distractions at home that reduce the concentration of attention in distance teaching are stronger and therefore face-to-face teaching is even more necessary especially in a subject with laboratory requirements such as science.

**Table 6.** Students' concentration during distance teaching compared to face-to-face teaching.

|  | Much Decreased | Decreased | Same | Increased | Much Increased |
|---|---|---|---|---|---|
| Primary education | 6.3% | 35.7% | 48.1% | 7.2% | 2.7% |
| Secondary education | 31.0% | 27.4% | 28.9% | 9.6% | 3.1% |

**Table 7.** Conceptualizing physics lesson content during distance teaching compared to in-person teaching.

|  | Much Worse | Worse | Same | Better | Much Better |
|---|---|---|---|---|---|
| Primary education | 5.3% | 38.1% | 48.1% | 6.3% | 2.2% |
| Secondary education | 21.5% | 35.7% | 30.9% | 9.2% | 2.7% |

On one hand, students' low concentration can be attributed to conditions existing in the home environment [14–16] accompanied by teacher's weakness at enforcing discipline and providing motivation at a distance [12]. On the other hand, "science can't be learned by reading and discussion only" [12], while hands-on activities decreased, were eliminated, or were replaced [30,31,33].

Answers in Table 6 are fully compatible with those in Table 7, in which students' views on the conceptualization of science content during distance science teaching is presented. Most students in both educational stages claimed that they had poor cognitive achievement. Particularly, conceptualization in secondary education was poorer (57.2% if "much worse" and "worse" is added) (M = 2.36, Mdn = 2, SD = 0.81) than in primary education (43.4% if "much worse" and "worse" is added) (M = 2.63, Mdn = 3, SD = 0.78), $t(576) = 1.96$, $p < 0.05$. These differences may be attributed to the difference in concentration and in the depth of the concepts taught, as in secondary education the concepts are given and analyzed in a formal and mathematical way, which makes it difficult for a considerable number of students to respond.

The finding of poor scientific knowledge as a consequence of science teaching at a distance is also included in similar studies. Leonardi et al. [35] found "gaps in students' knowledge and investigation skills" at a rate of 67% (p. 4). Wisanti et al. [12] found it difficult to explain scientific concepts, because "practical work . . . that supports students' understanding about concept of science . . . became main problem of teacher" (p. 6) during remote teaching.

### 4.5. Regarding Overall Evaluation of Science Teaching at a Distance

In Table 8, students' answers regarding the overall evaluation of science teaching at a distance are presented. In both levels, students' answers show that most of them did not prefer or enjoy remote teaching in science classes. Secondary education students

seem to be much less satisfied by science distance teaching related to face-to-face teaching (50.3% if "much less attractive" and "attractive" is added) than primary students (27.5%). Findings reveal that science distance education satisfied more primary students, with possible explanations being the greater variation in teaching style, less workload, slightly higher concentration of attention, and slightly better understanding of concepts compared to secondary students.

**Table 8.** Overall evaluation of science teaching at a distance related to in-person teaching.

|  | Much Less Attractive | Less Attractive | The Same | More Attractive | Much More Attractive |
|---|---|---|---|---|---|
| Primary education | 8.7% | 18.8% | 39.7% | 25.1% | 7.7% |
| Secondary education | 25.9% | 24.4% | 29.4% | 12.7% | 7.6% |

The general preference for face-to-face science teaching has also emerged in studies by other researchers of science education. In the UK, more than 70% of teachers thought that distance teaching had a negative impact on students, especially on lower-attaining students, on those with special educational needs and disabilities, and on economically disadvantaged students [35], (p. 41–43). In UAE, most of the science teachers (46%) prefer teaching via a traditional classroom [31], (p. 114), while parents in Turkey found "face-to-face education at schools more effective" [34], (p. 1907).

Table 9 presents students' answers in the question "what did you like most during science teaching at a distance," after indexing and categorizing. Primary and secondary students gave almost the same answers. A significant amount of both primary (28.80%) and secondary students (36.2%) answered that they liked nothing in the science distance teaching, revealing a general frustration, a finding which is in line with recent literature [39]. However, it seems that the big gain from distance education is the use of audiovisual material that may not have been utilized either in primary education (32.5%) or in secondary education (38.2%) in face-to-face teaching. It appears that teachers have improved their digital knowledge and skills, as inferred from the use of a stylus or other technology tools during the lesson, according to student responses and similar research in the field [12,32]. Small facilities, such as the comfort of the home environment, the short duration of the teaching hour, and the non-requirement of a mask, were pointed out by 22.3% of primary students and 13.1% of secondary students. Finally, it is considered noteworthy that 6.3% of primary students and 8.5% of secondary students evaluated positively the attitudes of teachers during science distance teaching, who provided increased leniency to the students, while also making an obvious effort to respond to the difficult requirements of the science distance teaching. We consider that the similarities of the answers given by the students in both levels support the reliability of these findings and give a representative picture of the conditions of the implementation of the science distance teaching in Greece.

**Table 9.** What did you like most during science teaching at a distance?

|  | Primary Education | Secondary Education |
|---|---|---|
| Audiovisual material (video, power point, digital stylus pen, etc.) | 32.5% | 38.2% |
| Nothing | 28.8% | 36.2% |
| Several facilities (shorter lesson hour, no face mask needed, the comfort of home, no school commuting) | 22.3% | 13.1% |
| Teachers' effort to cope with the situation | 6.3% | 8.5% |
| Email communication with teachers | 4.8% | 2.5% |
| Other | 5.3% | 1.5% |

The increase of digital instructional tools was clearly reported by teachers in many other studies on distance science education. Simulations, animations, and digital applications were commonly utilized by teachers for science teaching at a distance [32,33]. Al Darayseh [31] quotes an extended table with 29 different websites, software, and applications used for teaching science (p. 113). Besides students, some parents also found that "visuals and videos used were arresting" [34], (p. 1907). In the UK, a free digital resource for science teaching named Explorify was used by 81% of teachers for activities in the classroom, by 32% as a toolkit for teaching science, and by 25% as a toolkit for leading science [35], (p. 44).

Table 10 presents students' answers in the question "what did you not like during science teaching at a distance" as they were formatted after indexing and categorization. Both primary and secondary students gave similar answers. Technical issues, such as poor internet connection, seemed to be the major issue during the implementation of science distance teaching, both in primary (33.8%) and secondary (23.1%) education, which is in accordance with research in other countries [1,2,22]. However, primary students' younger age, lack of familiarity with technology, and the potential need for parental help with networking probably explain the difference between stages. The lack of student–student and student–teacher face-to-face interaction was pointed out by 27.7% of primary students and by 18.1% of secondary students in agreement with the corresponding closed-ended question but also with the literature [17,19,40]. In addition, the lack of experiments was mentioned by 11.9% of primary students and by 8.5% of secondary students as has been pointed out in other studies [29,31].

**Table 10.** What did you not like during science teaching at a distance?

| | Primary Education | Secondary Education |
| --- | --- | --- |
| Technical issues (poor internet connection, crowded frequency channel, etc) | 33.8% | 23.1% |
| Weak communication between classmates and teachers | 27.7% | 18.1% |
| Lack of experiments | 11.9% | 8.5% |
| I did not like anything | 5.5% | 13.6% |
| Weak concentration (very boring, sitting all day in front of a laptop, headaches, etc) | 6.6% | 10.1% |
| Ineffective teaching just to cover the material | 5.3% | 12.6% |
| Several issues (way of examination, way some classmates were constantly noisy, too much homework, etc.) | 4.2% | 9.0% |
| I liked everything | 4.4% | 4.5% |
| No answer | 0.6% | 0.5% |

Although the above worries and concerns were pointed out in higher percentages by primary students, there were negative aspects of science distance teaching that seem to bother secondary students slightly more, such as ineffective teaching, the inability to concentrate in class, and several procedural issues. The same tendency prevailed among the students who had an overall negative image of science distance teaching, answering "I didn't like anything" by 13.6% in the secondary education against 5.5% in primary education. These responses are compatible with those of the closed-ended questions that refer to changes in the teachers' teaching practices, the assignment of homework, the concentration of students' attention, and the understanding of concepts.

Our findings are compatible with those of other researchers of science education on two main problems of remote teaching. Wisanti et al. [12] concluded that technical issues are the greatest difficulty in online learning. Besides internet access, "the lack of application operating skill by teacher (45.45%) and the lack of students' communication equipment

(42.27%)" was recorded (p. 4). According to Leonardi et al. [35], both students and teachers struggled with accessibility to equipment and suitability of devices. In addition, "a lack of internet access or a reliable Wi-Fi connection means that online learning simply is not possible for some families" [35], (p.12).

The greatest difficulty in distance science teaching was to teach without experimentation, as reported by many researchers. McPherson and Pearce [30] found it difficult to teach an online science class, when students had no opportunities "to interact and engage with experiments, demonstrations and modelling scientific concepts" (p. 8). Thinking about assigning experiments for home, Leonardi et al. [35] concluded that "teachers felt that they could not expect too much of parents in terms of the equipment or time required for an experiment" (p. 39). So "46% of science teachers are missing the hands-on activity and experiential learning as well" [31], (p. 114). Consequently, the majority of teachers have mainly used online simulations, interactive games, and reading material to replace investigations, group work, and analyzing data [29,35].

## 5. Conclusions

This study aimed at mapping the similarities and differences between the views of students in Y6 and students in Y10 on science distance education during its implementation due to the COVID-19 pandemic in the school year 2020–2021 in Greece. The findings revealed that both primary and secondary students had mainly negative views on science distance teaching. Their views seemed to be negatively influenced by the technical issues such as poor network connection, the limited student–student and student–teacher interactions, and by the limited or complete lack of laboratory activities. However, a limited number of students pointed out some positive experiences from the prolonged period of distance education. Such positive aspects are the increased and effective spread of audiovisual material, such as selected software, animations, and videos. This kind of material helped in the conceptualization of the content under consideration and are recommended to be adopted in science teaching more broadly, and not only in the context of distance education. Regarding the comparison between the stages, it seems that more technical problems were reported by primary education students, while secondary education students reported more profound worries and difficulties related to teaching practices.

Students' answers can be used for the appropriate planning of successful science distance education at both stages. For example, the use of audiovisual material and applications (simulations, ppt, video experiments, etc.) can improve students' understanding of concepts. In contrast, elements that should be avoided are technical problems, limited communication, and lack of experiments. These results are consistent with relevant research on distance education both during closure due to the COVID-19 pandemic [13] and earlier [41].

Limitations arise from a limited and convenient sample and do not allow trustworthy generalizations. The greatest risk of bias is that students who had good access to an internet connection and equipment may have responded to our survey and students with less facilities may not to have responded.

Findings call for further research, including science teachers' perspectives on distance education, and particularly, the pedagogical, didactical, and institutional aspects that distance education affects. Moreover, different socio-cultural backgrounds need to be investigated. This is in line with the OECD report for Greece, which reveals poor availability of technology and technical assistant staff in schools [26]. Students' opinions could be checked by mapping teachers' corresponding opinions [42–46] as teachers were the persons who undertook and implemented the unprepared transition to distance education during the COVID-19 pandemic. Finally, the present study asks for further research so as to raise the quality of science teaching material for online settings, along with teachers' training in both digital technologies and distance education, in order for distance teaching to become more effective not only in fighting of the pandemic but in other circumstances as well.

**Author Contributions:** Conceptualization, C.S. and A.M.; methodology, C.S. and A.M.; investigation, C.S. and A.M.; writing—original draft preparation, C.S. and A.M.; writing—review and editing, C.S. and A.M. All authors have read and agreed to the published version of the manuscript.

**Funding:** This research received no external funding.

**Institutional Review Board Statement:** Ethical review and approval were waived for this study since during the design and implementation procedures were followed to ensure ethical standards regarding the anonymity of the participants, parents' consent and password protected data storage.

**Informed Consent Statement:** Informed consent was obtained from all subjects involved in the study.

**Data Availability Statement:** Data sharing not available.

**Conflicts of Interest:** The authors declare no conflict of interest.

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
