# Peer review of "Science Teaching at a Distance in Greece: Students’ Views"

_education, doi:10.3390/educsci13040395_

Round 1

Reviewer 1 Report

In the 30th row: „that has not been investigated yet” – I think the spectrum of empirical research findings are very wide. Several articles, research reports have described this shift from different aspects (didactic, the attitude of students, the attitude of teachers etc.). Later the author(s) wrote that there are a lot of researches in this field (row 54) – this is a contradiction in my opinion.

The literature review contains the notion of distant education and online learning, the level of digital skills and some findings from Greece. Because the central issue is the discipline of “science” and the research question refers to views about science distant education of different age group perhaps these elements have to be the most significant parts of the literature review. There are a lot of articles which reveal the features of science teaching during pandemic situation. So I have that feeling that the issue of the analysis and the literature review are not fitting to each other. 

Author(s) have to explain why this research question was chosen, and research question has to be embedded in literature review. We have to know why is so important to give an answer of this research question. And if we have already known the answer what are the consequences? Are there any practical implications according to the age differences?

I think author(s) should formulated more research questions (and not only one) and hypotheses too (according to the questionnaire).

The description of sample method is a little bit vague for me. I can not see the description of school selection (Why these schools were selected? From how many schools are children coming? Was there used any sample method? Do these schools have common features – e.g. according to the parental background?)

The description of the sample is missing (sex, type of settlement, SES etc.). If these elements were covered by the questionnaire the authors can use these as independent variables which can form the attitude and experiences of students.

I think the quantitative part is a little bit modest. We can not see any statistical tests – only percentages (chi square test, variance analysis, correlations may be used). I think the authors may have more possibilities if they use this database. The analysis of open-ended questions and the system of categories seem to be more interesting. Perhaps the authors can make a connection between these categories and the independent variables (sex, parental background etc. if they were covered) and/or the other part of the questionnaire.

The analysis and the conclusion have to be more thorough – with more citations. Less descriptive and more interpretative elements are required.

Table 10. – a lots of ‘space’ are missing the table

To sum up, in my opinion the literature review has to be improved, the research questions/hypotheses have to be reformulated, description of sample method and sample have to be broaden, new elements have to be incorporated in the quantitative part and conclusion has to be improve too.

Reviewer 2 Report

The article is interesting as it relates to the emergence of online education in schools at specific schools in Greece.

There are minor spelling and/or grammatical errors visible see page 2 line 24 Quit and Quiet, for example.

The terms K-6 and K-10 are understandable and confusing. it may be best to state "students in Y6, and Y10"

Concern the sample size does not report the age group. Also concerning is that there was no ethics indication for surveying participants under 18 years old. The authors must indicate if they received Ethical Approval, and supply the code, as well as enumerate Gender differences and age groups in percentages.

Limitations should be at the end of the article and provide a few limitation examples, such as the data analysis process which seems to be descriptive only.

The data analysis could go beyond reporting differences between Primary and Secondary students. For example the authors could attempt to report on the preferred pattern of learning for all Primary students and then contrast it to Secondary.

There is no Discussion section that compares the findings to the literature explanations. Yet the article provides a few opportunities to discuss findings with support from the literature.

The conclusion is primarily an opinion section. 

What is meant by learning online? There is no statistics provided regarding students access to computer and smartphones at home (online learning) (See Gromik and Litz, 2021 for example: reporting on technology access of Emirati students.). The authors could access online data to report technology access by Greek students. 

Sincerely

Round 2

Reviewer 1 Report

All of my suggestions were accomplished by the author(s).